Research

 

**Subject Area:**
biophysics/developmental biology

zebrafish, early development, strong static magnetic field, microtubules, spindle, mitosis

**Authors for correspondence:**
Hongyuan Jiang
e-mail: jianghy@ustc.edu.cn
Bing Hu
e-mail: bhu@ustc.edu.cn

# Strong static magnetic field delayed the early development of zebrafish

Shuchao Ge[1], Jingchen Li[2], Dengfeng Huang[1], Yuan Cai[1], Jun Fang[3], Hongyuan Jiang[2] and Bing Hu[1]

[1]Hefei National Laboratory for Physical Sciences at the Microscale, CAS Key Laboratory of Brain Function and Disease, School of Life Sciences, and [2]CAS Key Laboratory of Mechanical Behavior and Design of Materials, Hefei National Laboratory for Physical Science at the Microscale, Department of Modern Mechanics, CAS Center for Excellence in Complex System Mechanics, University of Science and Technology of China, Hefei, Anhui 230027, People's Republic of China
[3]Anhui Province Key Laboratory of Condensed Matter Physics at Extreme Conditions, High Magnetic Field Laboratory of the Chinese Academy of Science, Hefei, Anhui 230031, People's Republic of China

SG, 0000-0002-6590-9050; YC, 0000-0003-0089-6911; HJ, 0000-0003-0266-3011; BH, 0000-0001-8627-5272

One of the major topics in magnetobiology is the biological effects of strong static magnetic field (SMF) on living organisms. However, there has been a paucity of the comprehensive study of the long-term effects of strong SMF on an animal's development. Here, we explored this question with zebrafish, an excellent model organism for developmental study. In our research, zebrafish eggs, just after fertilization, were exposed to a 9.0 T SMF for 24 h, the critical period of post-fertilization development from cleavage to segmentation. The effects of strong SMF exposure on the following developmental progress of zebrafish were studied until 6 days post-fertilization (dpf). Results showed that 9.0 T SMF exposure did not influence the survival or the general developmental scenario of zebrafish embryos. However, it slowed down the developmental pace of the whole animal, and the late developers would catch up with their control peers after the SMF was removed. We proposed a mechanical model and deduced that the development delaying effect was caused by the interference of SMF in microtubule and spindle positioning during mitosis, especially in early cleavages. Our research data provide insights into how strong SMF influences the developing organisms through basic physical interactions with intracellular macromolecules.

## 1. Introduction

In modern societies, compared with geomagnetic field (GMF), people have access to much stronger static magnetic fields (SMFs), like the one used in magnetic resonance imaging (MRI) and those manufactured in several national high magnetic field laboratories. Taking advantage of the large equipment, biological scientists have studied the effects of strong SMF on various living organisms from multiple aspects (as recently reviewed in [1]). However, few have systematically studied the question during very early development. Most previous reports have focused on only a few aspects or a very short time during development.

In addition, researchers have observed multifarious and sometimes contradictory results. While some reported that strong SMF did not exert severe effects on the development of *Xenopus laevis* (6.34 T for 6 and 18 h or 8 T for 20 h) [2–4] or mice (1.5 and 7 T, 75 min each day during the entire pregnancy, or 4.7 T exposure from 7.5 to 9.5 day of gestation) [5,6], others observed obvious side effects, including the altered cleavage plane (1.7–16.7 T exposure from fertilization to the third cleavage) [7,8] or cortical pigmentation (9.4 T exposure from 15 to 109 min) [9] in *Xenopus* eggs, retarded development and aberrant gene expression in *Xenopus* embryos (15 T exposure from uncleaved to 2-cell, 2-cell to blastula and blastula to neurula) [10], shortened lifespan in *Caenorhabditis elegans* (8 T for 1, 3 and 5 h) [11], delayed hatching in mosquito eggs (9.4 and 14.1 T exposure for

70–163 h) [12], reduced viability in mouse fetuses (1.5 T exposure for 30 min) [13] and so on. These studies provided valuable information about the effects of strong SMF on development. However, they only observed a few aspects, or were restricted to either immediate or postnatal effects. A full and comprehensive view is still lacking as to the effects of strong SMF on early development.

To study the long-term effects of strong SMF on early development from multiple dimensions, we chose an aquatic model organism, *Danio rerio* (zebrafish), which has never been used in previous reports on this question. Compared with reported animals, zebrafish possesses several outstanding advantages. First, zebrafish develops faster than *Xenopus* and mice. Starting from a zygote, it completed cleavage, blastula, gastrula and segmentation stages in 24 h. After the pharyngula period, zebrafish starts hatching at 48 h post-fertilization (hpf) and reaches early larval period at 72 hpf [14]. Such a fast development allows tracking from fertilization to larvae in only one week. Second, distinct from the *in utero* development of mice, the *in vitro* development avoids the interference from the female and allows manipulations of both control and experiment groups with intact embryos of the same batch (i.e. descendants of the same parents). Along with the transparency of the embryos and early larvae, it also facilitates detailed observation of the developmental process. Third, the mature behavioural testing methods of zebrafish [15] allow us to inspect the question from a functional aspect. Taken together, using zebrafish, we can obtain a relatively full view of the long-term effects of strong SMF on early development in a reasonably short period.

In our study, we exposed zebrafish eggs to 9.0 T SMF starting just after fertilization, and found that SMF did not affect the survival or malformation rate of embryos. Instead, it delayed the early development of the whole animal, as demonstrated by slower hatching, pharyngeal development and body growth, altered expression of indicator genes during development, and worse performance than control in visual function tests. However, the delaying effect of strong SMF was not permanent, since the embryos exposed to SMF would soon catch up with their control counterparts once returned to the normal condition. To explain the phenomena, we proposed a mechanical model that the strong SMF interfered with and lengthened the spindle positioning process by influencing the polymerization rate and inducing rotation and deformation of microtubules. Our simulation results indicated that the increment in the relaxation time of the positioning and orientation processes of spindle accumulated, making the effects of strong SMF become perceptible as developmental delay. We studied the long-term effects of strong SMF on the early development of zebrafish from multiple aspects, and provide a reasonable explanation of the results with biophysical methods.

## 2. Results

### 2.1. Strong SMF did not affect the survival or the general development scenario of zebrafish embryos

The eggs of wild-type zebrafish line AB were exposed to 9.0 T SMF from fertilization to 4, 8 or 24 hpf. When the SMF exposure finished, we checked the morphology of embryos,

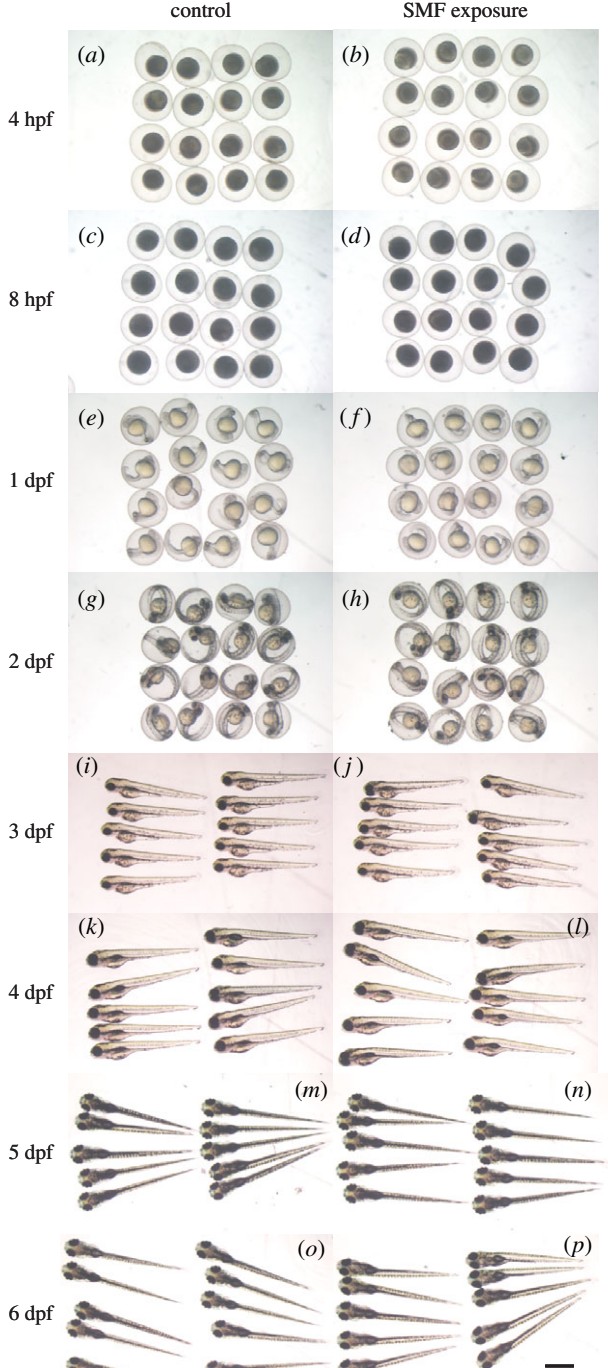

**Figure 1.** Snapshots of zebrafish embryos of both control and strong SMF-exposed groups at 4, 8 hpf and 1–6 dpf. (*a,c,e,g,i,k,m,o*) Morphology of control embryos; (*b,d,f,h,j,l,n,p*) morphology of SMF-exposed embryos. Note that before 3 dpf, zebrafish curled up in the chorion and looked spherical under a stereoscope. At 3 dpf, zebrafish hatched and stretched their bodies. Scale bar, 1 mm.

fixed (for exposure to 4 and 8 hpf) or alive (1 to 6 days post-fertilization (dpf)). Results showed that 9.0 T SMF did not affect the normal morphology of zebrafish embryos at 4 hpf (figure 1*a,b*), 8 hpf (figure 1*c,d*) and 24 hpf (figure 1*e, f*). After 24 h SMF exposure, the survival rate showed no significant difference with control (figure 2*a*).

In the following 5 days, we continued to snapshot the embryos to obtain a general view of the dynamic developing progress. After 24 h SMF exposure, embryos developed (figure 1*h,j,l,n,p*) in the same way as embryos in the control group (figure 1*g,i,k,m,o*), displaying no abnormalities with

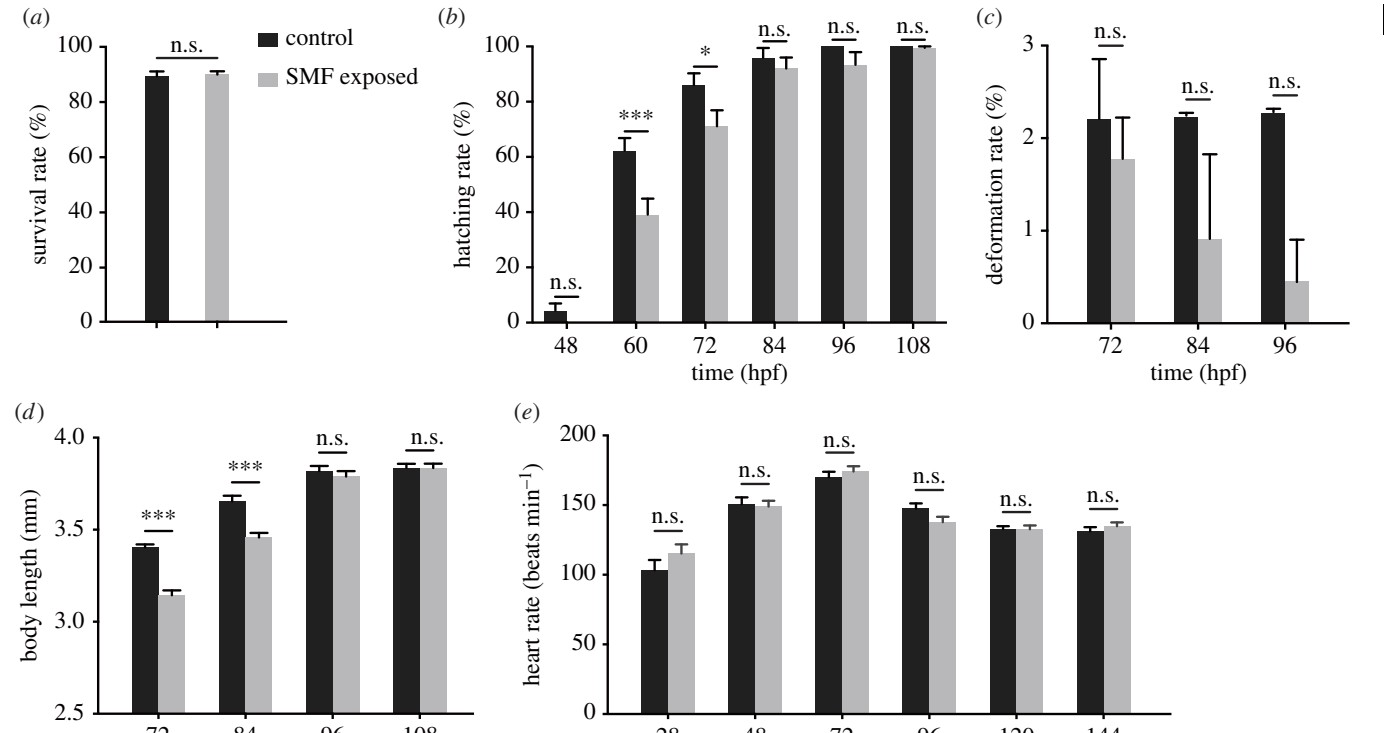

**Figure 2.** Developmental indices of zebrafish. (*a*) Survival rate of embryos with or without 24 h SMF exposure, calculated when embryos were fetched out from SMF. Data came from three samples with 50 embryos each (*p* = 0.77). (*b*) Hatching rate from 2 to 4.5 dpf, calculated every 0.5 day. Control data came from three samples with 91 embryos each, and SMF-exposed data were from three samples with 75 embryos each. From left to right, the *p*-value is 0.41, less than 0.001, 0.009, 0.46, 0.22 and 0.93. (*c*) Deformation rate calculated after most embryos had hatching. Data source was the same as *b*. From left to right, the *p*-value is 0.57, 0.10 and 0.03. (*d*) Body length measured with IMAGEJ. Data were collected from 30 samples. From left to right, the *p*-value is less than 0.001, less than 0.001, 0.43 and 0.99. (*e*) Heart rate from 1 to 6 dpf. Each data point came from 12 embryos. From left to right, the *p*-value is 0.06, 0.79, 0.52, 0.11, greater than 0.99, 0.54. Data are shown as mean with s.e.m. \**p* < 0.05, \*\**p* < 0.01, \*\*\**p* < 0.001, n.s., no significance.

optical checking. These preliminary observations showed that 9.0 T SMF did not influence the survival rate or the post-exposure development of zebrafish embryos.

## 2.2. Strong SMF temporarily delayed hatching, body growing and pharyngeal development of zebrafish embryos

To acquire more information, we monitored several developmental and physiological indices, where the effects of SMF surfaced. During development, zebrafish have to hatch from a layer of chorion that envelopes the inner embryo. As shown in figure 1*g,h,i,j*, zebrafish usually hatch from 2 to 3 dpf. So, from 2 dpf, we recorded the hatching rate every 12 h. Analysis showed clearly that at 2.5 and 3 dpf, significantly fewer embryos had hatched in the SMF-exposed group, compared with control. At 3.5 dpf, no significant difference in hatching rate was observed and most embryos had hatched in both groups (figure 2*b*). Thus, strong SMF delayed the hatching of zebrafish embryos.

After hatching, the deformation rate and body length of embryos were recorded. Although malformation happened in both groups, the rate was not significantly different (figure 2*c*). Consistent with the late hatching mentioned above, embryos exposed to SMF grew slower than the control, but catched up at 4 dpf (figure 2*d*). However, heart rate demonstrated no significant difference between SMF-exposed and control embryos (figure 2*e*), indicating that SMF did not influence heart development.

The seven pharyngeal arches (i.e. the Meckel's cartilage, the ceratohyal bone and the five pairs of ceratobranchial bones; figure 3*e'*) develop in a prototypic way and have long been used as an indicator of zebrafish development [16]. Results showed that at 4 dpf, the control embryo has fully developed the seven arches (figure 3*a,a'*), while SMF-exposed embryo has formed only three pairs of ceratobranchial bones (figure 3*b,b'*). After 24 h, it caught up with the control and grew the remaining two pairs of ceratobranchial bones at 5 dpf (figure 3*c,c',d,d'*). At 6 dpf, the pharyngeal arches showed the same morphology between control and SMF-exposed embryos (figure 3*e,e',f,f'*). So, although the strong SMF temporarily delayed the development of the pharyngeal arches, the embryos would soon catch up with the control.

Taking all the developmental indices into consideration, we concluded that 9.0 T SMF was neither lethal nor teratogenic to zebrafish embryos. It influenced multiple developmental aspects to varying extent (i.e. delaying the hatching, body growing and pharyngeal development, but not affecting the heartbeat of the embryos). Exposed embryos would catch up with the control.

## 2.3. Strong SMF altered the expression of indicator genes during early development

After observing the morphological indices of the delaying effect of SMF, we asked how it was reflected at the molecular level. We chose several indicator genes during early

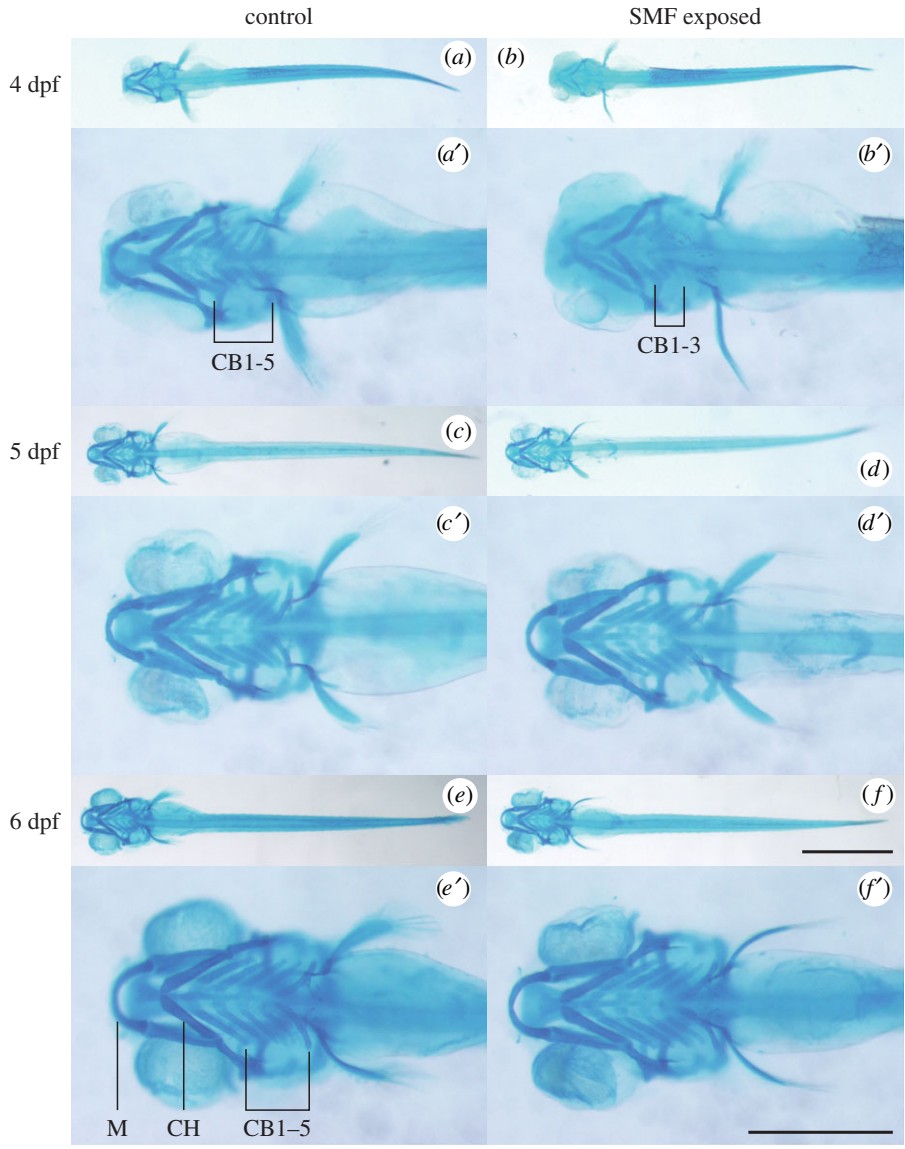

**Figure 3.** Alcian blue staining of the seven pharyngeal arches of zebrafish embryos from 4 to 6 dpf. (*a,a′*) Control embryo formed five ceratobranchial (CB) bones at 4 dpf. (*b,b′*) SMF-exposed embryo developed only three CB bones. (*c,c′,d,d′*) Pharyngeal arches of 5 dpf control (*c,c′*) and SMF-exposed (*d,d′*) embryos displayed no significant difference. (*e,e′,f,f′*) Pharyngeal arches of 6 dpf control (*e,e′*) and SMF-exposed (*f,f′*) embryos displayed no significant difference. The seven pharyngeal arches are Meckel's cartilage (M), ceratohyal (CH) bone and the five pairs of ceratobranchial (CB1–5) bones as illustrated in *e′*. (*a–f*) use the same scale bar, 1 mm. (*a′–f′*) are the magnifications, respectively, and use the same scale bar, 0.5 mm.

development of zebrafish and performed real-time quantitative RT–PCR to assess the gene expression level. In order to obtain a long-term cognition of the effects of SMF, we collect samples consecutively from 1 to 6 dpf.

Among the eight genes analysed, *mylz3* and *pvalb* indicate the development of fish skeleton [17] and both of them showed significantly lower expression level at 1 dpf in SMF-exposed embryos than control (figure 4*a,b*). The difference disappeared in the following days except at 5 dpf for *pvalb*, indicating that embryos in both groups were growing body. Hence, strong SMF exposure delayed the myogenesis of zebrafish embryos, but the gap would soon be evened up.

*syn2a* is regarded as a marker of synapse formation in the nervous system [18,19]. Its expression showed no significant difference between the two groups except at 3 dpf, when SMF-exposed embryos expressed higher level of *syn2a* than control (figure 4*c*). A similar case happened to *nestin*, an intermediate filament protein gene and a marker of the neural stem and progenitor cells [20,21].

At 2 dpf, significantly more *nestin* was expressed in SMF-exposed embryos than control (figure 4*d*). These results indicate that SMF-exposed embryos were trying to catch up with control.

*elavl3* (encoding HuC), *gfap* (astrocyte marker) and *mbp* (oligodendrocytes and myelin marker) are expressed exclusively in the nervous system [18]. Although *gfap* showed no significant difference between the two groups at 1–6 dpf (figure 4*f*), less *elavl3* (1 dpf, figure 4*e*) and *mbp* (3 and 4 dpf, figure 4*g*) was expressed in SMF-exposed embryos than control, indicating a slower development of the neural system. *shha* is expressed in developing neurons and other tissues [18]. Similar to *gfap*, its expression showed no significant difference between the two groups (figure 4*h*).

The molecular analysis above demonstrated that strong SMF exposure affected the expression of different genes to various extent. Considering the roles of the marker genes during development, the key events of myogenesis, neural system development and synaptogenesis were all affected in the strong SMF-exposed embryos.

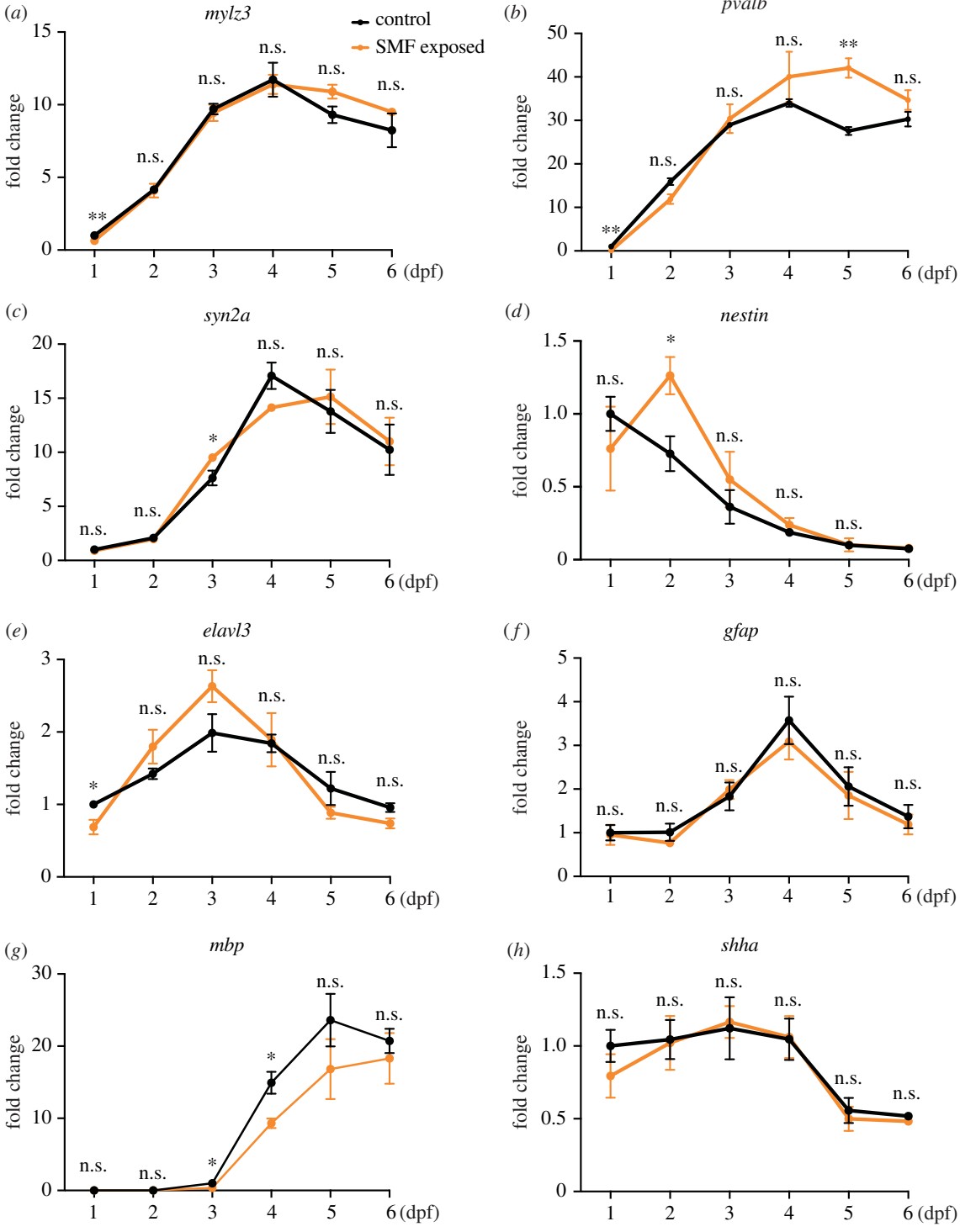

**Figure 4.** qRT–PCR analysis of eight indicator genes from 1 to 6 dpf. Each data point came from three samples, with 30 embryos each at 1 and 2 dpf, and 15 embryos each at 3–6 dpf. Results were analysed with multiple *t*-tests and the *p*-values in each figure, from left to right, are (*a*) 0.008, 0.90, 0.71, 0.82, 0.10, 0.35; (*b*) 0.002, 0.04, 0.68, 0.36, 0.004, 0.19; (*c*) 0.63, 0.74, 0.05, 0.08, 0.69, 0.82; (*d*) 0.49, 0.04, 0.45, 0.36, 0.95, 0.79; (*e*) 0.04, 0.20, 0.13, 0.90, 0.24, 0.08; (*f*) 0.88, 0.30, 0.70, 0.52, 0.78, 0.63; (*g*) 0.49, 0.62, 0.04, 0.03, 0.29, 0.57; and (*h*) 0.33, 0.93, 0.87, 0.94, 0.66, 0.11. Data were normalized and shown as mean with s.e.m. *$p < 0.05$, **$p < 0.01$, ***$p < 0.001$, n.s., no significance.

## 2.4. Strong SMF did not affect the swimming motion, but delayed the development of optokinetic response in zebrafish larva

After checking the marker genes that played important roles in muscular and neural development of zebrafish, we wondered how it reflected on function (i.e. the effects of strong SMF on animal behaviour).

First, we recorded the free swimming behaviour of zebrafish after hatching (figure 5*a*). At 3 dpf, larvae had just hatched from the chorion and barely moved. They became more active at 4 dpf, and swam for more time and a greater distance. However, neither the moving time nor the average speed displayed significant difference between SMF-exposed and control groups. (Figure 5*b,c*). Therefore, strong SMF did not influence the free swimming behaviour of zebrafish.

Next, we used a more refined behavioural test, optokinetic response (OKR), to assess the visual and neuromuscular functions of zebrafish larvae. In this test, the body of the larva was stabilized by a viscose liquid (methylcellulose) and was surrounded by light and dark gratings. When the gratings

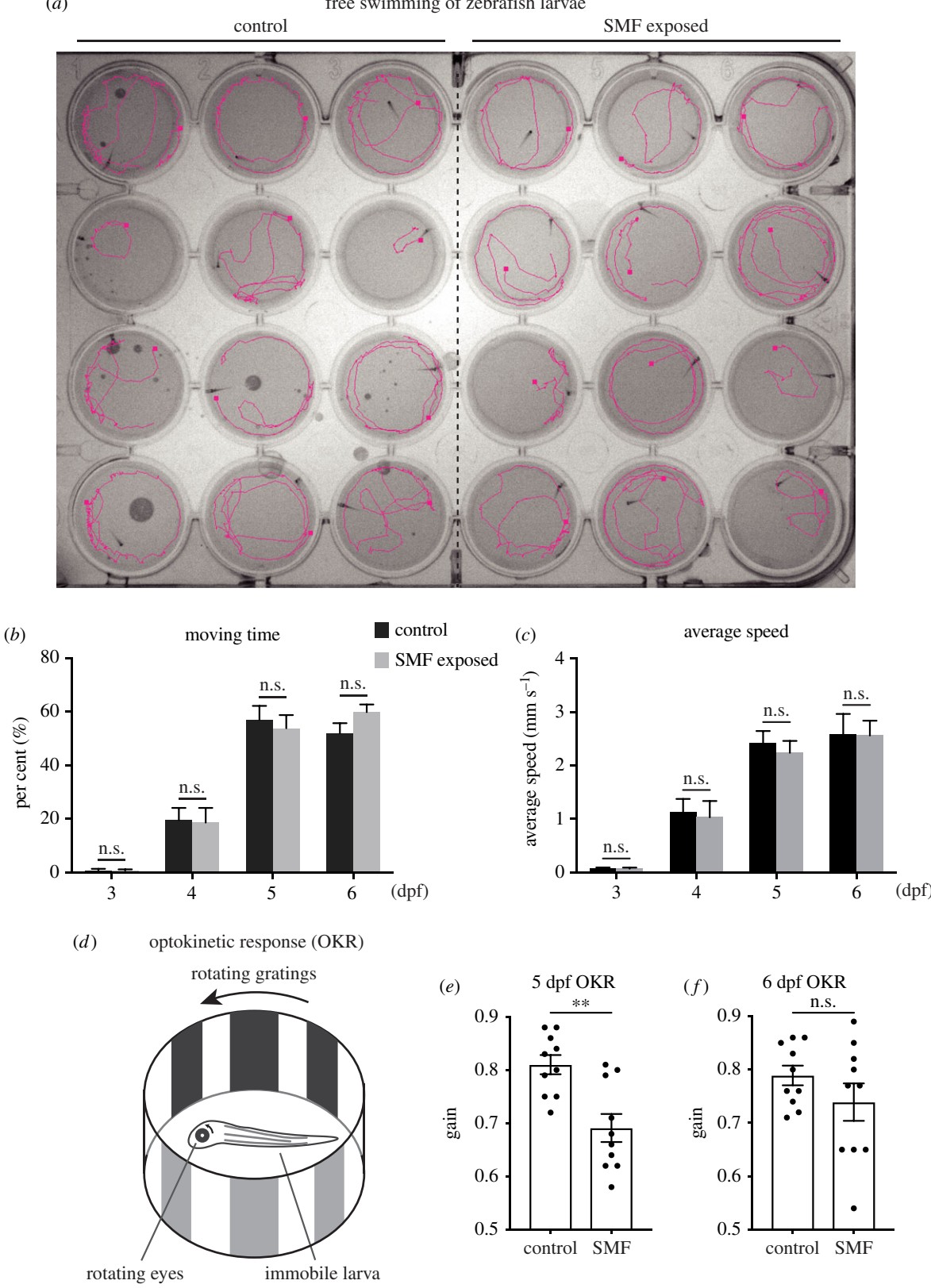

**Figure 5.** Behavioural tests of the effects of strong SMF. (*a*) The magenta line portrayed the swimming path of zebrafish larvae in 1 min. Experiments were performed in a 24-hole dish, with one larva to each hole. (*b*) Moving time of larvae in free swimming tests. (*c*) Average speed of larvae in free swimming tests. (*d*) Schematic depiction of OKR tests. (*e,f*) Performance of larvae in OKR tests at 5 and 6 dpf, respectively. Each data point came from 24 larvae in (*b*, *c*). Data were analysed with multiple *t*-tests and the *p*-values are (*b*) 0.96, 0.90, 0.67, 0.10; (*c*) 0.91, 0.79, 0.57, 0.96. (*e*) and (*f*) used 10 larvae in each group. Data were analysed with *t*-test and the *p*-values are (*e*) 0.002 and (*f*) 0.22. All data are shown as mean with s.e.m. *$p < 0.05$, **$p < 0.01$, ***$p < 0.001$, n.s., no significance.

began to rotate, the larva would try to track the gratings and its eyes would rotate (figure 5*d*). Gain, the ratio of the velocity of larvae eye and the rotating light source, was used to measure the larva's performance. It denoted how well the eye can

follow the rotating grating pattern. At 5 dpf, embryos exposed to SMF scored significantly worse than the control (figure 5*e*). One day later, they caught up and showed the same gain as control (figure 5*f*). Since the performance of

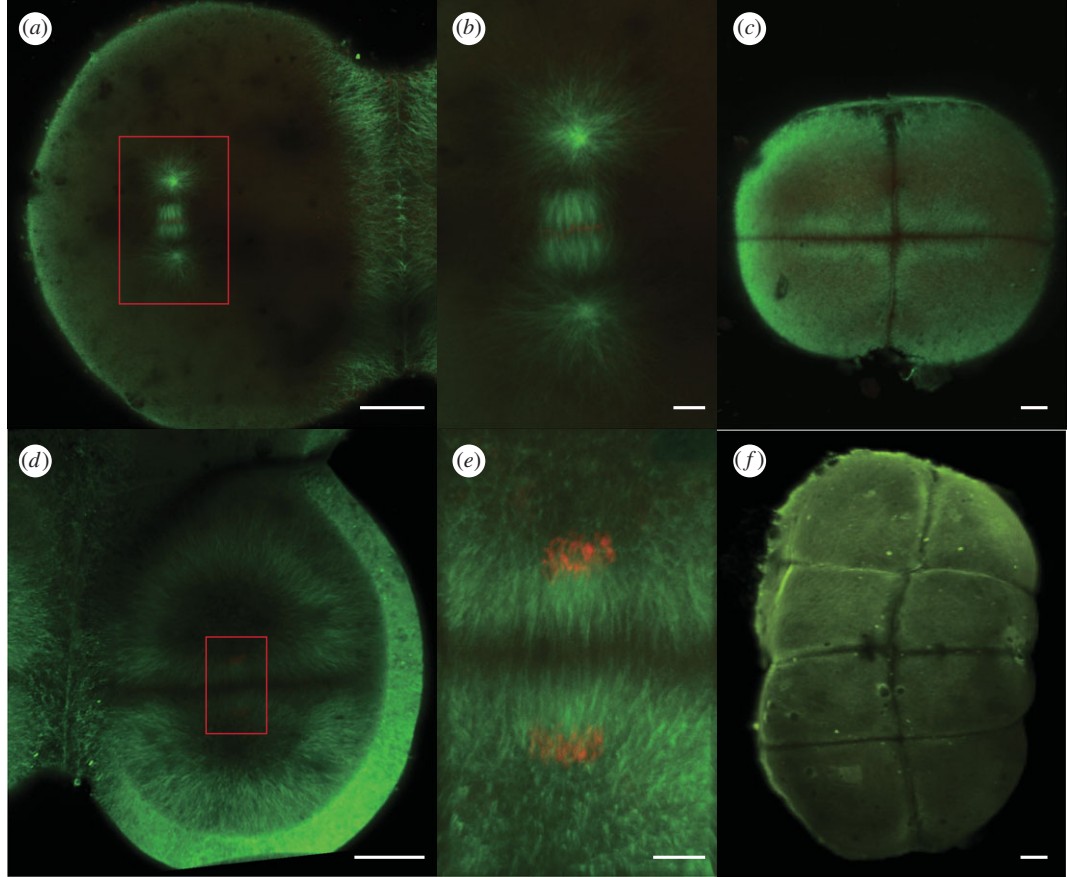

**Figure 6.** Immunohistochemistry of zebrafish embryos during early cleavage in control condition. (*a*) Metaphase of the second cleavage, (*b*) magnification of the spindle in *a*, (*c*) embryo of 4-cell stage after the second cleavage, (*d*) telophase of the third cleavage, (*e*) magnification of the departing spindle in (*d*) and (*f*) embryo of 8-cell stage after the third cleavage and before the telophase of the fourth cleavage. Green indicates microtubules and red indicates chromosomes. Scale bar in *a*, *c*, *d* and *f* is 50 μm. Scale bar in *b* and *e* is 10 μm.

OKR depends on the maturation extent of the retina [22,23], these results indicated that strong SMF delayed the development of retina.

Both the free swimming and OKR behavioural assay showed that although strong SMF did not influence the simple motion, it delayed the development of elaborate visual motor function in zebrafish larvae. SMF-exposed embryos soon caught up with and behaved as well as the control.

## 2.5. Theoretical research indicated the effect of SMF on microtubules and spindle positioning delayed the early cleavages

When figuring out why strong SMF delayed the early development of zebrafish, we speculated that SMF must have influenced certain structures or processes that were pivotal to the normal progression of development. We focused on spindles and explored how the effect of strong SMF on mitotic spindles led to developmental delay of the whole animal.

### 2.5.1. Basis of theoretical model: diamagnetic anisotropy of microtubules and unique features of zebrafish early development

Microtubules are composed of α- and β-tubulin dimers, whose diamagnetic anisotropy makes them align along the weakest diamagnetic axis (i.e. the α–β axis) under the

magnetic torque exerted by strong SMF. This, in turn, endows the microtubules with a tendency of aligning parallel with SMF direction [24]. In dividing cells, the spindle is composed of a large amount of microtubules, and is also affected by strong SMF, as verified by cellular experiments [25,26].

Moreover, a unique feature of zebrafish is that during cleavage, the cells are quite large, with a size of hundreds of micrometres (figure 6*a,c,d,f*). In that case, the size of the spindle reaches an upper limit (figure 6*b,e*) [27,28], and the astral microtubules span the large cell to position and orient the spindle (figure 6*a,d*). Since the relaxation time of positioning and orienting spindle to the cell centre increases exponentially with the cell size [25,29], it is a huge project for microtubules during zebrafish cleavages. Any disturbances from exterior factors, like strong SMF, will greatly influence the normal progression of mitoses.

In addition, after fertilization, zebrafish starts the first cleavage in about 30–45 min. Then, the embryo synchronously divides every 15 min in the following 8 cell cycles, until the midblastula transition period [30]. Such a mitotic synchrony leaves a long time window for the effects of strong SMF.

Taking into consideration the large spindle and synchronous cleavage of zebrafish, we hypothesized that due to the magnetic torque from strong SMF, microtubules needed more time to correctly position and orient the spindle to the cell centre, thus lengthened the cell cycle. The effect accumulated and finally became perceptible as developmental delay of the whole embryo. We will explain and simulate the process in detail in the following model.

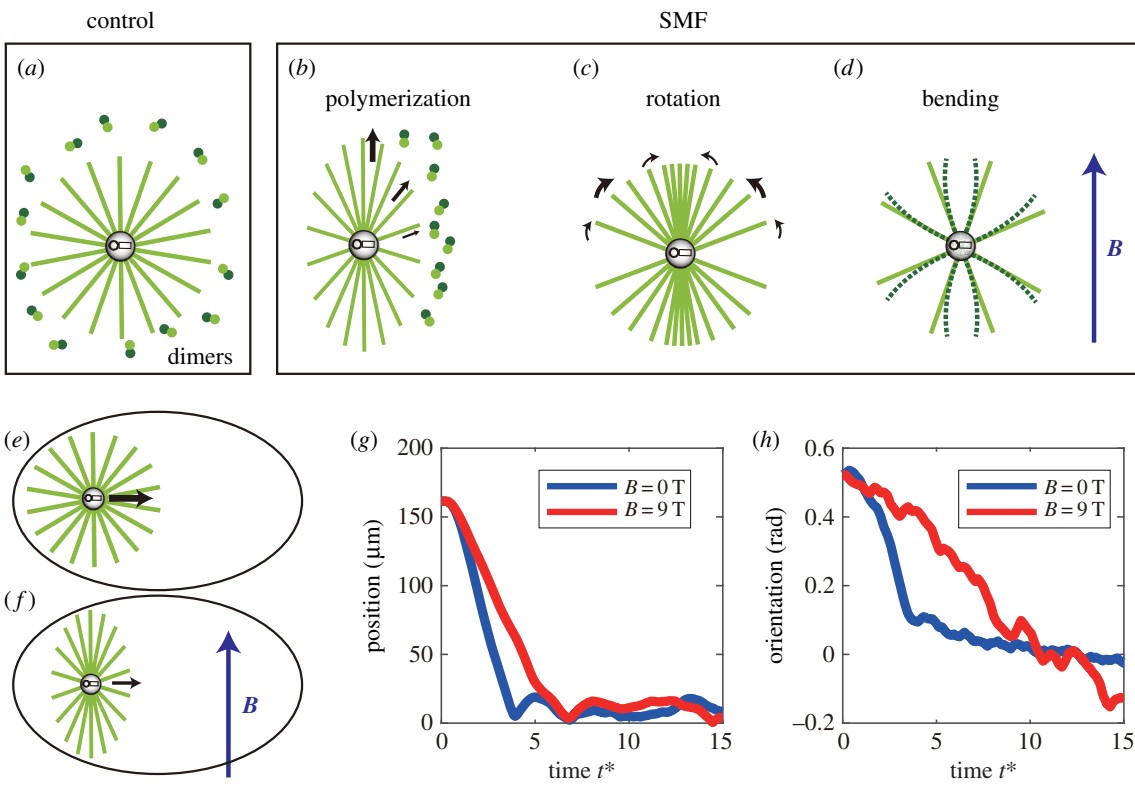

**Figure 7.** Model of the delaying effect of strong SMF on early development. (*a*) Dissociative tubulin dimers are randomly distributed in cytoplasm in control condition. (*b*) Tubulin dimers tend to orient with $\alpha$–$\beta$ axis parallel to SMF direction, altering the polymerization rate at astral microtubule ends. The thickest arrow means the largest polymerization rate. (*c*) Microtubules are rotated under magnetic torque from SMF. The thickest arrow means the largest rotating speed. (*d*) Microtubules bend to the direction of SMF. The largest deflection happens at the microtubule ends. (*e*,*f*) Schematic diagram shows the positioning and orienting of spindle by microtubules (*e*) without or (*f*) with an external SMF. (*g*) Computational simulations of the positioning process of spindle with and without SMF. $t^* = t/\tau_d$, where $\tau_d$ is the relaxation time for positioning process without SMF [25]. (*h*) Computational simulations of the orientation process of spindle with and without SMF. $t^* = t/\tau_\alpha$, where $\tau_\alpha$ is the relaxation time for orientation process without SMF [25].

## 2.5.2. Theoretical model: strong SMF influenced the polymerization rate and induced rotation and deformation of microtubules

There is a dynamic equilibrium between microtubules and dissociative tubulin dimers in cytoplasm, and the end of microtubules undergoes constant assembly and disassembly [31]. So, we first investigated the effect of strong SMF on the polymerization of microtubules.

Without SMF, the dissociative tubulin dimers are randomly oriented in cytoplasm (figure 7*a*). However, they experienced magnetic torque and took a preferential direction under SMF, which will influence the polymerization rate at microtubule ends (figure 7*b*).

For one dimer, the magnetic torque is

$$M_0 = \frac{\Delta\chi B^2}{2\mu_0}\sin 2\theta, \tag{2.1}$$

where $\Delta\chi$ is the magnetizability difference between the orthogonal axes of the maximal diamagnetic anisotropy, and $\Delta\chi = 1.243 \times 10^{-32}$ m$^3$ [24]. $B$ is the magnetic strength, i.e. $B = 9T$, $\mu_0$ is the permeability of vacuum and $\theta$ is the angle between the dimer axis and SMF direction.

Under SMF, the distribution of dissociative dimers follows the Boltzmann distribution

$$P = k_0 e^{-\frac{E}{k_B T}}, \tag{2.2}$$

where $k_0$ is the normalized coefficient, $k_B$ is the Boltzmann constant, $T$ is the absolute temperature and $E$ is the energy required for resisting the magnetic torque in the direction $\theta$, such that

$$E = \frac{\Delta\chi B^2}{2\mu_0}\sin^2\theta. \tag{2.3}$$

Therefore, in SMF, the polymerization rate of microtubules in the direction $\theta$ is

$$v(\theta) = v_0 e^{-(\Delta\chi B^2 \sin^2\theta/2\mu_0 k_b T)}, \tag{2.4}$$

where $v_0$ is the polymerization rate without SMF and estimated as $v_0 = 0.125 \ \mu$m s$^{-1}$ [32].

Equation (2.4) shows that the polymerization rate is the same as the rate without SMF, if the microtubule grows parallel to the direction of SMF. Otherwise, the rate is decreased by SMF. When the microtubule is perpendicular to SMF, the polymerization rate is the smallest. This mechanism reveals that the astral microtubules prefer to polymerize along SMF (figure 7*b*).

Next, we consider the effect of SMF on microtubules. Most microtubules have 13 monofilaments, and the number of dimers in a piece of microtubule with the length of $L$ is $13L/d_0$, where $d_0 = 8$ nm is the length of dimer. Therefore, the magnetic torque on the microtubule is

$$M = \frac{13L\Delta\chi B^2}{2d_0\mu_0}\sin 2\theta. \tag{2.5}$$

royalsocietypublishing.org/journal/rsob    Open Biol. **9**: 190137

The rotating speed is

$$\phi = \frac{2M}{\xi L^2},\tag{2.6}$$

where $\xi$ is the drag coefficient of unit length of microtubule and estimated as $\xi = 1$ pNs nm$^{-1}$ [32]. Therefore, microtubules tend to rotate to the direction of SMF, and when $\theta = \pi/4$, the rotating speed is the largest (figure 7c).

In equation (2.5), we treat the microtubule as a rigid pole and derive the magnetic toque on it. However, the torque is distributed along the microtubule, which is

$$M(x) = \frac{13\Delta\chi B^2}{2d_0\mu_0}(L - x)\sin 2\theta,\tag{2.7}$$

where $0 \leq x \leq L$ represents the distance to the negative end of the microtubule.

Based on the small deformation assumption, the deflection of microtubules is

$$\omega = \frac{13\Delta\chi B^2 \sin 2\theta}{2d_0\mu_0\mathrm{EI}}\left[\frac{1}{2}Lx^2 - \frac{1}{6}x^3\right],\tag{2.8}$$

where $\mathrm{EI} = 33.12$ pN m$^2$ is the bending rigidity of the microtubule [33].

The maximal deflection $\omega_{\max}$ is at the end of the microtubule (i.e. $x = L$). The result is that microtubules bend to the direction of SMF (figure 7d).

### 2.5.3. Computational simulation: effects of strong SMF on microtubules interfered with the positioning and orientation of spindle and prolonged the cell cycle

The analysis above demonstrated that compared with control, microtubules in strong SMF are influenced in three aspects (i.e. the altered polymerization rate, rotation and bending). How are these effects translated to developmental delay of the whole animal? The reason is that microtubules modulate the oscillatory behaviour of spindles, whose correct and accurate positioning and orientation are vital to the normal progression of cell cycle [34–36].

In large cells, the positioning and orientation of spindles are driven by the pulling force along the astral microtubules (figure 7e,f) [27]. This process is interfered with by strong SMF, as verified in experiment performed on Xenopus eggs [7]. In the normal condition, the pulling force is distributed along the microtubules (figure 7e), but it has a deviation when the microtubule is bent by SMF (figure 7f). In this case, the pulling force decreases as

$$f_c^* = f_c\sqrt{L^2 - \omega_{\max}^2},\tag{2.9}$$

where $f_c$ is the pulling force on the microtubule without bending. Obviously, the pulling force is weaker in strong SMF, which means microtubules cannot work as efficiently as they do in control to position and orient the spindle.

To obtain a quantitative recognition, we used the computational model developed in our recent work [25,32] to simulate the positioning and orientation of spindles under an external strong SMF. We made three assumptions here. First, microtubules are nucleated on the two spindle poles, grow radially and display the dynamic instability [37–39]. Second, microtubules experience pulling force proportional to their length from cytoplasmic motors [27,40]. Third, microtubules reaching the cortex are given a pushing force by

polymerization, and a pulling force by bounding of cortical dynein [41,42]. Together these forces position the spindle to the cell centre and orient it along the long axis of the cell.

After adding the influence of SMF on microtubules (i.e. the altered polymerization rate, rotation and deformation), we found microtubule bending plays a major role in the process. As expected, the spindle was positioned (figure 7g) and oriented (figure 7h) more slowly than control in large cells, especially during the first cleavage when the zebrafish egg was about 700 µm. Thus, more time was needed in one cell cycle and after lots of cycles, the accumulation of the extra time was finally demonstrated as developmental delay of the whole animal.

## 3. Conclusion

Exposure to 9.0 T SMF from fertilization to 24 hpf was neither lethal nor teratogenic to zebrafish. The strong SMF temporarily delayed the developmental pace of the whole animal, as indicated by slower hatching, pharyngeal development and body growth, altered gene expression and worse performance in behaviour tests, compared with control. However, the late developers would catch up with their control peers when the strong SMF was removed. We proposed a model in which the strong SMF affected the polymerization rate and induced rotation and bending of microtubules. This reduced the force and efficiency of microtubules when completing the task of positioning and orienting the spindle, leading to longer cell cycles of cleavages. The cumulative effects finally manifested as delayed development of the whole animal. The computational simulation well explained our experiment results.

## 4. Discussion

### 4.1. Compare the effects of strong SMF on development between zebrafish and other organisms

We found both consistencies and contradictions between ours and previous results. We did observe hatching delay of zebrafish embryos, as reported in mosquito eggs [12], but we did not find malformation induced by SMF exposure, which was reported on Xenopus [10]. Multiple factors may account for such variability. First, the SMF applied differed in strength, orientation and homogeneity. SMF up to 8 T did not influence the development of Xenopus eggs [2]. However, 15 T SMF retarded the development and induced malformations and aberrant gene expression in Xenopus [10]. Second, different organisms might respond variously to strong SMF due to their inherent characteristics. As mentioned above, while 8 T SMF declared no significant changes in Xenopus [2], 1 T was enough to induce abnormal mitotic spindle in cultured cells [43]. Third, treatments before SMF exposure also played a role. The reorganization of cortical pigmentation happened only when the jelly coat were removed from Xenopus eggs [9].

### 4.2. Compare the biophysical models of explaining the biological effects of strong SMF

As to the mechanism explanation, we proposed a model and performed computational simulation that targeted the positioning and orienting of spindle, while Valles Jr et al. [44]

provided a qualitative model that focused on the centrosome replication and spreading process. The reason was that we studied different aspects of the question. While Valles Jr and colleagues studied the immediate effects of strong SMF on cleavage in several hours [7,8], we observed the long-term effects of SMF on early development in 6 days. Both our models were based on the diamagnetic anisotropy of microtubules, but turned to different events of mitosis in order to explain the phenomena we observed in experiments.

While they share many similarities, *Xenopus* and zebrafish are inherently different during cleavage. Cleavage of *Xenopus* is holoblastic and mesolecithal (i.e. both the animal and vegetal poles divide and the yolk deposits moderately). Zebrafish, on the other hand, is meroblastic and telolecithal (i.e. the yolk concentrate densely in the vegetal pole and only the blastodisc divides [45]). As to the cleavage, *Xenopus* divides vertically in the first and second mitosis and horizontally the third, while zebrafish divides vertically until the sixth mitosis, the first horizontal one [46]. So, cleavage of *Xenopus* is complete while zebrafish is not, and the cleavage orientation is different.

All the differences above reminded us to be very wary of simply applying others' theory to interpret our results. In our view, both previous and our reports were based on experimental observations and studied the effects of strong SMF on development. However, when we scrutinize the organisms and conditions used, none of them were completely the same. The multifarious results might just demonstrate the variety and complexity of the effects of strong SMF.

### 4.3. Modelling the influence of SMF on mitotic spindle

Previous studies have shown that SMF can reorient spindle through deflecting microtubules, and thus can change the third cleavage orientation in *Xenopus* [7,8,44]. However, the models in these studies were qualitative and phenomenological. Based on the physical mechanism of the effect of SMF on microtubules [24], we considered as complete as possible a mechanism in the model, including the microtubule growing dynamic, deformation and reorientation. Each of them can only induce a little influence on microtubule, but together they can significantly delay the positioning of the spindle.

Other studies also showed SMF may influence chromosomes and other cytoskeleton [26,47], and thus influence spindle orientation and cell motion [26,48]. The effect of SMF on cellular physiological function should be implemented through a synthetical action on multiple intracellular structures or biomacromolecules. Therefore, modelling more mechanisms of them may be able to explain or predict more biological phenomena in SMF.

## 5. Material and methods

### 5.1. Zebrafish maintenance and eggs collection

Zebrafish were reared according to standard procedures under 28.5°C with 14 h/10 h light/dark cycle and fed brine shrimps twice a day [49]. To obtain eggs through natural mating, we put male and female adults in the same container separated by a plate. At dawn the next morning, we removed the plate and eggs were soon laid and fertilized. We reared eggs in Hank's solution.

Zebrafish larvae were anaesthetized with a solution of tricaine methane-sulfonate (MS-222, Sigma, St Louis, MO, USA), and all efforts were made to minimize suffering.

### 5.2. The strong SMF exposure

The strong SMF was provided by High Magnetic Field Laboratory of the Chinese Academy of Science. It is produced by a vertical superconducting magnet (American Magnetics, Inc., Oak Ridge, TN, USA). During the experiment, zebrafish embryos were first placed in a glass Petri dish, then put in a stainless iron bore (figure 8*a,b*). The iron bore was put in the centre of the superconducting magnet (figure 8*c*), where the magnetic field is homogeneous, and the strength is 9.0 T. The temperature was controlled at 28.5°C by water circulation. The ventilation inside the bore was maintained by an air pump. Samples were exposed to the strong SMF from fertilization to 24 hpf. To observe the morphology of early embryos, exposure starts from fertilization to 4 or 8 hpf. The control group was kept in an incubator with the same temperature and ventilation, except that it was exposed to the GMF.

### 5.3. Recording of heart rate of zebrafish embryos

The embryo's heartbeats in 30–60 s were counted and recorded under a stereoscope. The number of heartbeats was divided by the time, producing the heart rate in beats per minute.

### 5.4. Alcian blue staining of pharyngeal cartilage

We performed cartilage staining according to the protocol of Westerfield [49], as follows. Anaesthetize zebrafish embryos with tricaine and fix them in 4% paraformaldehyde. Dehydrate in 25%, 50%, 75% and 95% ethanol/PBS solution. Stain the pharyngeal cartilage overnight in room temperature with 0.1% alcian blue prepared with 80% ethanol/20% glacial acetic acid. Rehydrate with 95%, 75%, 50% and 25% ethanol solution. Digest with 0.05% trypsin prepared in saturated sodium tetraborate solution. Mix equal volume of 2% $H_2O_2$ and 1% KOH to prepare the bleaching solution. During bleaching, leave the cap of the tube open and observe every 15–30 min. The younger the embryo, the shorter the time needed to be bleached. In room temperature, 5 dpf embryos need 3–4 h to bleach.

### 5.5. Quantitative real-time RT–PCR

Gene expression was analysed with quantitative PCR, using primers (table 1) as the literature described [17,18]. Each sample was obtained from 15 or 30 embryos with or without SMF exposure.

### 5.6. Behavioural tests

The free swimming of zebrafish larvae was recorded and analysed by Noldus equipment and software (Noldus, Wageningen, The Netherlands). OKR was measured by the system developed by our own laboratory [50].

### 5.7. Spindle staining and imaging

Bright field images of the whole larvae were snapshot with a stereoscope (Olympus, Tokyo, Japan). Mitotic spindle was

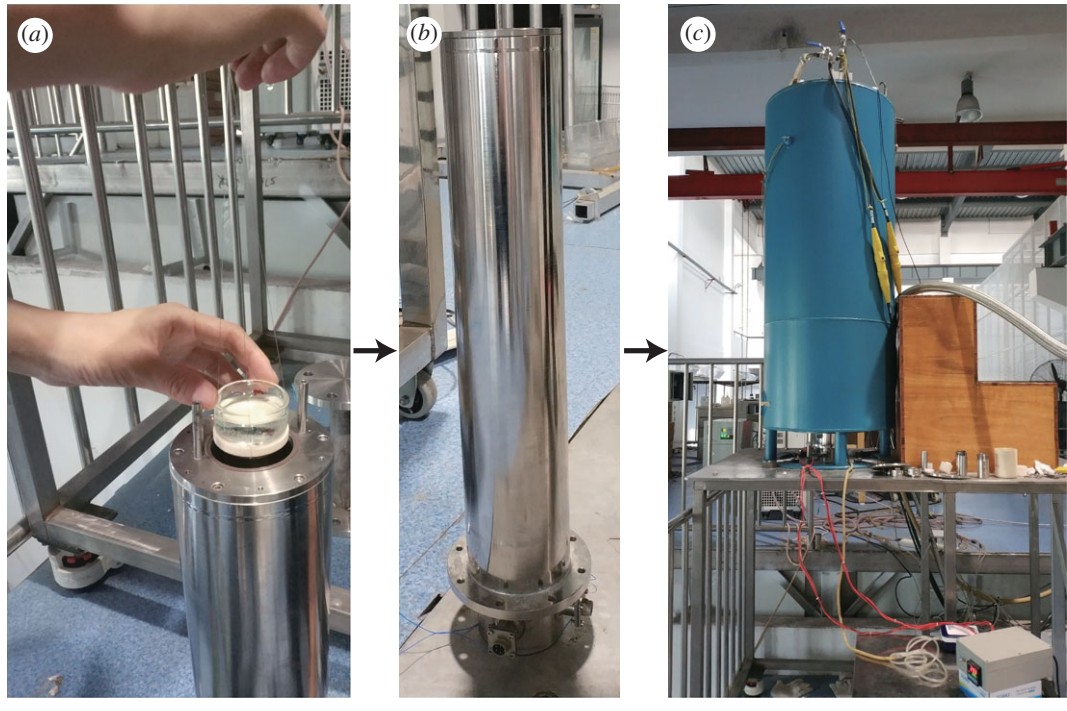

**Figure 8.** Exposing zebrafish embryos to strong SMF. (*a*) Zebrafish eggs were put in a Petri dish, which was placed in a stainless iron bore. (*b*) The stainless iron bore. (*c*) The iron bore was put into the centre of the superconducting magnet.

**Table 1.** Primers used in this paper.

| gene name | forward primer (5′ to 3′) | reverse primer (5′ to 3′) |
|---|---|---|
| *rpl13α* (internal control) | TCTGGAGGACTGTAAGAGGTATGC | AGACGCACAATCTTGAGAGCAG |
| *mylz3* | GTAAGAACCCCACCAACAAG | GGTTCGCTCATCTTCTCACC |
| *pvalb* | GGAATTCTCAAGGACGAGGA | TGCAGGAACAGTTTCAGCTC |
| *syn2a* | GTGACCATGCCAGCATTTC | TGGTTCTCCACTTTCACCTT |
| *nestin* | ATGCTGGAGAAACATGCCATGCAG | AGGGTGTTTACTTGGGCCTGAAGA |
| *elavl3* | AGACAAGATCACAGGCCAGAGCTT | TGGTCTGCAGTTTGAGACCGTTGA |
| *gfap* | GGATGCAGCCAATCGTAAT | TTCCAGGTCACAGGTCAG |
| *mbp* | AATCAGCAGGTTCTTCGGAGGAGA | AAGAAATGCACGACAGGGTTGACG |
| *shha* | GCAAGATAACGCGCAATTCGGAGA | TGCATCTCTGTGTCATGAGCCTGT |

stained according to previous procedures [51,52] with some modifications. All the Petri dishes and straws used here should be made of glass. Fertilized eggs during cleavage were first dechorionated and then fixed in phosphate-buffered saline (PBS) with 4% paraformaldehyde, 0.25% glutaraldehyde, 0.05% EGTA and 0.2% Triton X-100 at 4°C overnight. After washing, dehydrate in gradient methanol/PBS solution, i.e. 25, 50, 75 and 100% methanol. Leave the samples at −20°C overnight. Rehydrate with gradient methanol/PBS solution in reverse order. Treat with 0.5 mg ml$^{-1}$ NaBH$_4$ for 4 h to terminate the action of glutaraldehyde (Leave the caps open and perform in the fume hood). Remove half or more than half of the yolk for good permeation of the following reagents, especially antibody. Block at 4°C overnight with 10% bovine serum albumin (BSA), 5% normal goat serum in 0.2% PBS Triton. Next eggs were incubated in anti-α-tubulin antibody (monoclonal from rabbit, 1 : 2500, Cat. T-6074, Sigma,

St Louis, MO, USA) at 4°C for 3 days. Wash at room temperature for 8 h and apply the secondary antibody Alexa fluor goat anti-rabbit 488 (Invitrogen, Carlsbad, CA, USA) at 4°C overnight. Chromosomes were stained with propidium iodine (PI, 1:100 000) at room temperature for 30 min. Fluorescent images were obtained under a confocal microscope (Olympus, Tokyo, Japan) with 10× or 60× objectives.

## 5.8. Software and statistical analysis

Body length of zebrafish larvae were calibrated with ImageJ (NIH, Bethesda, MD, USA). Fluorescent images were processed with Imaris 7 (Oxford Instruments, Zurich, Switzerland). All data in this paper were analysed with *t*-test in Prism 8 (GraphPad Prism, San Diego, CA, USA) and displayed as mean with s.e.m. Significant difference was set as *$p < 0.05$, **$p < 0.01$ and ***$p < 0.001$.

## 5.9. Mechanical model and simulation method

The mechanical model was developed based on a general dynamic computational model for mitotic spindles. Considering the diamagnetism of microtubules, we simulated the dynamic positioning and orientation processes of the spindle in an elliptical cell (long axis 600 μm, short axis 300 μm, corresponding to the first cleavage) by programming a Monte Carlo algorithm in MATLAB [25,32].

Ethics. All animal manipulations were conducted in strict accordance with the guidelines and regulations set forth by the University of Science and Technology of China (USTC) Animal Resources Center and University Animal Care and Use Committee. The protocol was approved by the Committee on the Ethics of Animal Experiments of the USTC (Permit Number: USTCACUC1103013).

Data accessibility. This article has no additional data.

Competing interests. The authors declare no competing interests.

Funding. This study is supported by grants from Chinese MOST (grant no. 2019YFA0405600) and NSFC (grant nos U1332136 and 81790643).

Acknowledgements. A portion of this work was performed on the Steady High Magnetic Field Facilities (SM1 superconducting magnet), High Magnetic Field Laboratory, CAS. We thank Dr An Xu, Dr Qingyou Lu and Dr Ze Wang for sharing equipment and experiences about biological manipulations with strong SMF. We also thank Zhaoqiang Song for valuable suggestions on manuscript writing.

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
