## [Reviewer comments · Open Biology]

Review History

RSOB-19-0137.R0 (Original submission)

Review form: Reviewer 1

Recommendation

Accept with minor revision (please list in comments)

Do you have any ethical concerns with this paper?

No

Comments to the Author

This manuscript has used zebrafish as model to study the effect of strong magnetic field on development. It not only includes experimental results, but also has detailed theoretical calculations to explain their observed phenomenon. The results are very important for magneto-biology as well as the MRI community, especially for limiting pregnant women for high-field MRI exposure. I would like to recommend it for publication, after the following minor points are addressed.

1. The writing needs to be improved throughout the whole manuscript, including both grammar

and improper usage of some words.

2. As for the differences between different studies and their results, exposure time is also a very important factor, which should be discussed.
3. Abbreviations should be explained. Such as “dpf”.
4. The magnetic field exposure conditions need to be clearly stated. For example, what is the direction of the magnetic field? Is it upward or downward? What was the temperature inside the bore? Was the control group identically processed?
5. Why the visual motor functions were disturbed? Have similar results been reported before? What’s the potential mechanisms? The authors should add some discussion to this point.
6. Heart rate was not mentioned in the methods part.
7. “We should bear in mind that the ultimate object of gene expression was the normal morphology and function of the whole organism”. This sentence is weird.
8. Figure 4, why only control condition was examined?
9. “As expected, the spindle was positioned and oriented more slowly than control in large cells, especially during the first cleavage when the zebrafish egg was about 700 μ m (Fig.5 E and F).” Figure 5E and F have nothing to do with this statement. Please double check this.
10. Figure 4F, it is hard to tell whether it is at 8-cell stage based on this picture.

Review form: Reviewer 2

Recommendation

Reject – article is scientifically unsound

Do you have any ethical concerns with this paper?

No

Comments to the Author

The authors aim to explore the long-term effects of strong SMF on embryonic development using zebrafish as an animal model. Since the SMF majorly effects the cell cleavage and the early cell cleavage of zebrafish embryo is very fast (about 15 min for a cleavage division), it is reasonable that the strong SMF only have very mild effect on zebrafish development as shown by this study. In contrast, the human embryo undergoes very slow cleavage (7-8 cell at day 3), thus it is not suitable to use the fast-cleavage embryo model to conduct this type of study, in order to answer a human clinical question. Besides the major concern above, there are some major issues in this manuscript:

1. The authors did not show any picture of the embryos during the process of 24-hour SMF treatment while concluded that the strong SMF affected that cleavage and delayed the early development.
2. qPCR on three genes (Fig 2) is not adequate for the evaluation of delaying effect of SMF on molecular level.
3. Fig 4 is too descriptive rather than quantitative. Moreover, the Fig 4 did not show the SMF-treated group vs SMF-untreated group, which should give readers critical information about the SMF effects on cell cleavage and microtubule direction or other phenotypes.
4. The method about SMF treatment is very critical for this study, however the method is too brief.
5. Too many citations of previous studies and perspectives were presented in the Results part, which should mainly focus on the results of their own study.

Decision letter (RSOB-19-0137.R0)

30-Jul-2019

Dear Dr Ge,

We are writing to inform you that the Editor has reached a decision on your manuscript RSOB-19-0137 entitled "Strong Static Magnetic Field Delayed the Early Development of Zebrafish", submitted to Open Biology.

As you will see from the reviewers' comments below, there are a number of criticisms that prevent us from accepting your manuscript at this stage. The reviewers suggest, however, that a revised version could be acceptable, if you are able to address their concerns. If you think that you can deal satisfactorily with the reviewer's suggestions, we would be pleased to consider a revised manuscript.

The revision will be re-reviewed, where possible, by the original referees. You will need to address all comments from both reviewers. As such, please submit the revised version of your manuscript within four weeks. If you do not think you will be able to meet this date please let us know immediately.

When submitting your revised manuscript, please respond to the comments made by the referee(s) and upload a file "Response to Referees" in "Section 6 - File Upload". You can use this to document any changes you make to the original manuscript. In order to expedite the processing of the revised manuscript, please be as specific as possible in your response to the referee(s).

Please see our detailed instructions for revision requirements
<https://royalsociety.org/journals/authors/author-guidelines/>

Sincerely,
The Open Biology Team
mailto: openbiology@royalsociety.org

Associate Editor's comments:

The overall manuscript lacks rigorous evaluation on MRI effect on zebrafish embryos at both morphological and molecular levels. It remains unclear how fish study here is related to human patients in this regard.

Reviewer(s)' Comments to Author(s):

Referee: 1

Comments to the Author(s)

This manuscript has used zebrafish as model to study the effect of strong magnetic field on development. It not only includes experimental results, but also has detailed theoretical calculations to explain their observed phenomenon. The results are very important for magnetobiology as well as the MRI community, especially for limiting pregnant women for high-field MRI exposure. I would like to recommend it for publication, after the following minor points are addressed.

1. The writing needs to be improved throughout the whole manuscript, including both grammar and improper usage of some words.
2. As for the differences between different studies and their results, exposure time is also a very important factor, which should be discussed.
3. Abbreviations should be explained. Such as "dpf".
4. The magnetic field exposure conditions need to be clearly stated. For example, what is the direction of the magnetic field? Is it upward or downward? What was the temperature inside the bore? Was the control group identically processed?
5. Why the visual motor functions were disturbed? Have similar results been reported before? What's the potential mechanisms? The authors should add some discussion to this point.
6. Heart rate was not mentioned in the methods part.
7. "We should bear in mind that the ultimate object of gene expression was the normal morphology and function of the whole organism". This sentence is weird.
8. Figure 4, why only control condition was examined?
9. "As expected, the spindle was positioned and oriented more slowly than control in large cells, especially during the first cleavage when the zebrafish egg was about 700 μ m (Fig.5 E and F)." Figure 5E and F have nothing to do with this statement. Please double check this.
10. Figure 4F, it is hard to tell whether it is at 8-cell stage based on this picture.

Referee: 2

Comments to the Author(s)

The authors aim to explore the long-term effects of strong SMF on embryonic development using zebrafish as an animal model. Since the SMF majorly effects the cell cleavage and the early cell cleavage of zebrafish embryo is very fast (about 15 min for a cleavage division), it is reasonable that the strong SMF only have very mild effect on zebrafish development as shown by this study. In contrast, the human embryo undergoes very slow cleavage (7-8 cell at day 3), thus it is not suitable to use the fast-cleavage embryo model to conduct this type of study, in order to answer a human clinical question. Besides the major concern above, there are some major issues in this manuscript:

1. The authors did not show any picture of the embryos during the process of 24-hour SMF treatment while concluded that the strong SMF affected that cleavage and delayed the early development.
2. qPCR on three genes (Fig 2) is not adequate for the evaluation of delaying effect of SMF on molecular level.
3. Fig 4 is too descriptive rather than quantitative. Moreover, the Fig 4 did not show the SMF-treated group vs SMF-untreated group, which should give readers critical information about the SMF effects on cell cleavage and microtubule direction or other phenotypes.
4. The method about SMF treatment is very critical for this study, however the method is too brief.

5. Too many citations of previous studies and perspectives were presented in the Results part, which should mainly focus on the results of their own study.

Author's Response to Decision Letter for (RSOB-190137.R0)

See Appendix A.

RSOB-19-0137.R1 (Revision)

Review form: Reviewer 1

Recommendation

Accept as is

Do you have any ethical concerns with this paper?

No

Comments to the Author

No more questions.

Review form: Reviewer 2

Recommendation

Accept as is

Do you have any ethical concerns with this paper?

No

Comments to the Author

The revision has satisfied all my concerns.

Decision letter (RSOB-19-0137.R1)

08-Oct-2019

Dear Dr Ge,

We are pleased to inform you that your manuscript entitled "Strong Static Magnetic Field Delayed the Early Development of Zebrafish" has been accepted by the Editor for publication in Open Biology.

Article processing charge

Please note that the article processing charge is immediately payable. A separate email will be sent out shortly to confirm the charge due. The preferred payment method is by credit card; however, other payment options are available.

Sincerely,

The Open Biology Team
mailto: openbiology@royalsociety.org

Appendix A

Response to reviewers

Response to Associate Editor:

General comments:

The overall manuscript lacks rigorous evaluation on MRI effect on zebrafish embryos at both morphological and molecular levels. It remains unclear how fish study here is related to human patients in this regard.

Response:

We appreciate the associate editor's comments that point out the insufficiency of the original manuscript. We have performed extra experiments to reveal more details and robust evidences about the effects of strong SMF on morphological and molecular levels.

First, the morphology of zebrafish embryos was recorded at 4 and 8 hours after SMF exposure started, showing how zebrafish embryos were affected during strong SMF exposure. Results showed that strong SMF exposure had no severe harmful effects on zebrafish development.

Second, we used alcian blue to stain the pharyngeal arches of zebrafish embryos. The stereotyped development of the seven arches is used to evaluate the development of zebrafish [1, 2]. We compared the results of control and SMF exposed zebrafish, and found that strong SMF delayed the development of pharyngeal arches.

Third, we analyzed the expression of more marker genes, providing a broader view of the molecular effects of strong SMF on zebrafish development.

Last, the theme of the manuscript is studying the effects of strong static magnetic field (SMF) exposure on the development of zebrafish. This question has never been reported in previous literature. Considering the huge differences between zebrafish and human and the lack of original clinical data, we agree that it is inappropriate to extrapolate zebrafish study to human research. We deleted the contents related to human patients. Now the manuscript belongs to the field of basic biological study with an animal model. It reports the effects of strong SMF on zebrafish development and proposes a biophysical model to explain the phenomena.

Response to reviewer 1:**General comments:**

This manuscript has used zebrafish as model to study the effect of strong magnetic field on development. It not only includes experimental results, but also has detailed theoretical calculations to explain their observed phenomenon. The results are very important for magneto-biology as well as the MRI community, especially for limiting pregnant women for high-field MRI exposure. I would like to recommend it for publication, after the following minor points are addressed.

Response:

We greatly appreciate the comments from the reviewer. Our one-to-one response to the reviewer's questions can be found in the following.

Comment 1:

The writing needs to be improved throughout the whole manuscript, including both grammar and improper usage of some words.

Response 1:

We are thankful to the reviewer's suggestion. We have fully scrutinized the whole manuscript and made the following revisions.

First, we have deleted the redundant context related to human research and retained only the parts that are closely to the theme of the manuscript, i.e., the effects of strong static magnetic field (SMF) on the development of zebrafish.

Second, we have revised some sentences to ensure the grammar correctness of the manuscript.

Third, we have corrected the improper words to make the writing natural and fluid.

Last, based on new results obtained in the last weeks, we have enriched the results part and reorganized the discussion part. We believe these revisions will make the whole manuscript coherent and convincing.

Comment 2:

As for the differences between different studies and their results, exposure time is also a very important factor, which should be discussed.

Response 2:

We are thankful to the reviewer's suggestion. We have stated the exposure time when introducing previous studies.

Comment 3:

Abbreviations should be explained. Such as "dpf".

Response 3:

We are thankful to the reviewer's suggestion. We have added an "Abbreviations" part ahead of the "Introduction" part to make the paper more readable. Also, we provided the full name of the abbreviation the first time it appeared, such as "2 days post fertilization (2 dpf)".

Comment 4:

The magnetic field exposure conditions need to be clearly stated. For example, what is the direction of the magnetic field? Is it upward or downward? What was the temperature inside the bore? Was the control group identically processed?

Response 4:

We are thankful to the reviewer's comment. In "Material and methods" part, we have added the magnetic field conditions and showed how embryos were treated when exposed to the magnetic field in page 26-27 of the manuscript. Also, Figure 8 showed manipulation of SMF exposure.

Comment 5:

Why the visual motor functions were disturbed? Have similar results been reported before? What's the potential mechanisms? The authors should add some discussion to this point.

Response 5:

We are thankful to the reviewer's comment.

We have explained this result as the following:

Since the performance of OKR depends on the maturation extent of retina [3, 4], these results indicated that strong SMF delayed the development of retina. (page 12, line 221-

223.)

In detail, the optokinetic response (OKR) relies on the light perception and form vision of zebrafish visual system. Previous studies have demonstrated that the OKR appeared at 73 hours post fertilization (hpf), shortly after the retina began to function [5]. With the development of the zebrafish larva, especially the maturation of the retina, the OKR performance gradually increased from 73 to 96 hpf [4], and fully matured at 120 hpf, i.e., 5 days post fertilization (5 dpf) [3]. The OKR is a robust behavior pattern of zebrafish larvae and has long been used to evaluate the development of the zebrafish retina and visual function.

In our study, the strong magnetic field delayed the development of zebrafish, including the retina, so the performance of OKR was worse than the control at 5 dpf. However, as the SMF exposed zebrafish developed, the performance increased and showed no significant difference with the control at 6 dpf.

Comment 6:

Heart rate was not mentioned in the methods part.

Response 6:

We are thankful to the reviewer's comment. We have added the protocol of heart rate counting to the "Material and methods" part as the following:

Recording of heart rate of zebrafish embryos

The embryo's heartbeats in 30 to 60 seconds was counted and recorded under a stereoscope. The number of heartbeats was divided by the time, producing the heart rate in beats per minute. (page 27, line 466-469.)

Comment 7:

"We should bear in mind that the ultimate object of gene expression was the normal morphology and function of the whole organism". This sentence is weird.

Response 7:

We are thankful to the reviewer's comment. We have deleted this sentence and used this one to start the qPCR results:

After observing the morphological indices of the delaying effect of SMF, we asked how it was reflected on the molecular level. (page 10, line 164-165.)

Comment 8:

Figure 4, why only control condition was examined?

Response 8:

The purpose of figure 4 (figure 6 in revised manuscript) was to show the morphology, especially the large size of the spindle during zebrafish cleavage.

The lack of the SMF exposed results is due to the operation manner of the superconducting magnet. The voltage ramping process needs 43 minutes, i.e., the magnet needs 43 minutes to increase from 0 to 9 Tesla, and the same time to decrease from 9 to 0 Tesla.

During experiment, the samples were first placed into a bore that was made of stainless steel. Then the bore was put into the superconducting magnet before initiating voltage ramping. When magnetic field exposure ended, we took out the samples after the magnetic field decreased to 0 Tesla. We had no access to the samples when there were currents in the superconducting magnet. Also, we were suggested that we should not manipulate the samples inside the bore when the magnetic field was 9 Tesla.

However, if we chose to collect samples after the magnetic field decreased to 0 Tesla, it would miss the time window. The reason is that zebrafish eggs divide every 15 minutes during cleavage. For example, if we want to observe the second mitosis, we have to wait until the magnetic field decreases to 0 Tesla, which will take 43 minutes. When we finally take out the samples, the embryos have finished the second mitosis and may have begun the fourth or fifth division.

So, limited by the fast cleavage of zebrafish eggs and the relatively slow ramping process of the superconducting magnet, we are unable to provide microtubule staining of the SMF exposed group. Maybe other groups with alternative equipment, for example a ceramic bore or fast ramping magnet, can take out the samples without decreasing the magnetic field. In this case, the samples can be readily taken out of the bore while the field is unchanged.

Comment 9:

“As expected, the spindle was positioned and oriented more slowly than control in large cells, especially during the first cleavage when the zebrafish egg was about 700 μm (Fig.5 E and F).”

Figure 5E and F have nothing to do with this statement. Please double check this.

Response 9:

We are thankful to the reviewer’s comment. We have modified the sequence of the images in figure 5. Now we have changed this sentence to the following one:

As expected, the spindle was positioned (**Fig.7 G**) and oriented (**Fig.7 H**) more slowly than control in large cells,... (**page 19, line 359-361.**)

Comment 10:

Figure 4F, it is hard to tell whether it is at 8-cell stage based on this picture.

Response 10:

We are thankful to the reviewer’s comment.

Zebrafish eggs divide vertically from the first to the fifth cleavage, and the sixth cleavage was the first horizontal division [6]. In figure 4 F (figure 6 F in new manuscript), the blastomere is composed of eight cells arranged as a 4 \times 2 array. We can decide that the third cleavage has finished, but we do not know whether the fourth division has started. We can only decide that the fourth division has not entered the telophase, so we modified the caption of figure 4 F as the following:

F, Embryo of 8-cell stage after the 3rd cleavage **and before the telophase of the fourth cleavage.** (**page 36, line716.**)

Response to reviewer 2

General comments:

The authors aim to explore the long-term effects of strong SMF on embryonic development using zebrafish as an animal model. Since the SMF majorly effects the cell cleavage and the early cell cleavage of zebrafish embryo is very fast (about 15 min for a cleavage division), it is reasonable that the strong SMF only have very mild effect on zebrafish development as shown by this study. In contrast, the human embryo undergoes very slow cleavage (7-8 cell at day 3), thus it is not suitable to use the fast-cleavage embryo model to conduct this type of study, in order to answer a human clinical question. Besides the major concern above, there are some major issues in this manuscript.

Response:

We greatly appreciate the reviewer's comment. Considering the huge differences between the embryonic development of zebrafish and human and the lack of original clinical data, we agree that it is inappropriate to extrapolate zebrafish study to human research. We deleted the contents related to human patients. Now the manuscript belongs to the field of basic biological study with an animal model. It reports the effects of strong SMF on zebrafish development and proposes a biophysical model to explain the phenomena.

Our one-to-one response to the reviewer's questions can be found in the following.

Comment 1:

The authors did not show any picture of the embryos during the process of 24-hour SMF treatment while concluded that the strong SMF affected that cleavage and delayed the early development.

Response 1:

We are thankful to the reviewer's comment.

First, we have added pictures of embryos at 4 and 8 hours after SMF exposure begins. After comparing the developmental pace of control and SMF exposed embryos, we could see that SMF had no significant influence on the morphology of embryos.

Second, limited by the SMF condition, we are unable to take out the samples when the magnetic field is 9 Tesla. For example, if we want to observe the embryo during the second cleavage,

which happens about 60 minutes post fertilization, we have to wait until the magnetic field decreases to 0 Tesla, which will take 43 minutes. When we finally take out the samples, the right time window has been missed, and the embryos have finished the second cleavage and maybe have begun the fourth or fifth division. So we could not precisely provide the pictures of the first several cleavages during SMF exposure.

Third, we concluded by experimental phenomena that the strong SMF delayed the early development of zebrafish. When exploring the mechanism, based on our observation and previous literature, we turned to methods of modern mechanics and proposed a model focusing on the interaction of SMF on microtubules. Since we haven't provided the microtubule staining picture of SMF exposed embryos, we have strictly treated our theoretic model as a hypothesis instead of theorem. Considering the consistency of our results and those obtained on other animals, we are confident that the model is reasonable and worth of study.

Comment 2:

qPCR on three genes (Fig 2) is not adequate for the evaluation of delaying effect of SMF on molecular level.

Response 2:

We are thankful to the reviewer's comment. We have repeated the qPCR experiment and analyzed five more markers genes during zebrafish development, providing a broader view of the molecular effects of strong SMF. We also rewrote the qPCR results part as shown in page 10-11, line 163-198. A new figure (figure 4 in new manuscript) has been made to show the results.

Comment 3:

Fig 4 is too descriptive rather than quantitative. Moreover, the Fig 4 did not show the SMF-treated group vs SMF-untreated group, which should give readers critical information about the SMF effects on cell cleavage and microtubule direction or other phenotypes.

Response 3:

We are thankful to the reviewer's comment.

First, as we responded under comment 1, the lack of the SMF exposed results is due to the operation manner of the superconducting magnet. The voltage ramping process needs 43 minutes, i.e., the magnet needs 43 minutes to increase from 0 to 9 Tesla, and the same time to decrease from 9 to 0 Tesla.

During experiment, the samples were first placed into a bore that was made of stainless steel. Then the bore was put into the superconducting magnet before voltage ramping started. When magnetic field exposure ended, we took out the samples after the magnetic field decreased to 0 Tesla. We had no access to the samples when there were currents in the superconducting magnet. Second, we were suggested that we should not manipulate the samples inside the bore when the magnetic field was 9 Tesla.

Third, if we chose to collect samples after the magnetic field decreased to 0 Tesla, it would miss the time window. The reason is that zebrafish eggs divides every 15 minutes during cleavage. The fast cleavage makes it impossible to take out or observe the samples immediately. For example, if we want to observe the second mitosis, we have to wait until the magnetic field decreases to 0 Tesla, which will take 45 minutes. When we finally take out the samples, the embryos have finished the second mitosis and maybe have begun the fourth or fifth division. So, limited by the fast cleavage of zebrafish eggs and the relatively slow ramping process of the superconducting magnet, we are unable to provide microtubule staining of the SMF exposed group. Maybe other research groups with alternative equipment, for example a ceramic bore or a fast ramping magnet, can take out the samples without decreasing the magnetic field. In this case, the samples can be readily taken out of the bore while the field is unchanged.

In our manuscript, the purpose of figure 4 was to show the morphology, especially the large size of the spindle during zebrafish cleavage. The biological basis of the theoretic model was reasonable and solid. Since we have not obtained the microtubule staining of SMF exposed eggs, we can only state our model as a hypothesis instead of theorem or conclusion.

Comment 4:

The method about SMF treatment is very critical for this study, however the method is too brief.

Response 4:

We are thankful to the reviewer's comment.

We have added a figure and a paragraph in the “Methods and material” part to explain the SMF exposure procedure as the following:

The strong SMF exposure

The strong SMF is provided by High Magnetic Field Laboratory of the Chinese Academy of Science. It is produced by a vertical superconducting magnet (American Magnetics, Inc., Oak Ridge, Tennessee). During experiment, zebrafish embryos were first placed in a glass petri dish, then put to a stainless iron bore (Fig.8 A, B). The iron bore was put to the center of the superconducting magnet (Fig.8 C), where the magnetic field is homogeneous, and the strength is 9.0 T. The temperature was controlled at 28.5 °C by water circulation. The ventilation inside the bore was maintained by an air pump. Samples were exposed to the strong SMF from fertilization to 24 hpf. To observe the morphology of early embryos, exposure starts from fertilization to 4 or 8 hpf. The control group was kept in an incubator with the same temperature and ventilation, except that it was exposed to the geomagnetic field. (page 26-27, line 453-465.)

Comment 5:

Too many citations of previous studies and perspectives were presented in the Results part, which should mainly focus on the results of their own study.

Response 5:

We are thankful to the reviewer’s comment. The purpose of previous studies in the results part has two main purposes. One is to compare ours with others and the other is to provide basis of our theoretical model.

As to the former one, we have moved these comparisons to the Discussion part. Now in Results, we have focused on explaining our own observations.

As to the latter one, we must refer to previous work to show that the basis of our model has been verified by others and is solid. It's necessary to clarify the basic physical properties of microtubules and spindles before deducting the biophysical model. So, we have kept this part in the theoretic analysis.

1 Kimmel, C. B., Miller, C. T., Kruze, G., Ullmann, B., BreMiller, R. A., Larison, K. D., Snyder, H. C. 1998 The shaping of pharyngeal cartilages during early development of the zebrafish. *Developmental biology*. **203**, 245-263.

2 Schilling, T. F., Kimmel, C. B. 1997 Musculoskeletal patterning in the pharyngeal segments of the zebrafish embryo. *Development*. **124**, 2945-2960.

3 Huang, Y.-Y., Neuhauss, S. 2008 The optokinetic response in zebrafish and its applications. *Front Biosci*. **13**, 1899-1916.

4 Easter Jr, S. S., Nicola, G. N. 1997 The development of eye movements in the zebrafish (*Danio rerio*). *Developmental Psychobiology: The Journal of the International Society for Developmental Psychobiology*. **31**, 267-276.

5 Neuhauss, S. C. 2003 Behavioral genetic approaches to visual system development and function in zebrafish. *Journal of neurobiology*. **54**, 148-160.

6 Hisaoka, K. K., Battle, H. I. 1958 The normal developmental stages of the zebrafish, *Brachydanio rerio* (Hamilton-Buchanan). *Journal of Morphology*. **102**, 311-327.